

# Three-dimensional variational assimilation with a multivariate background error covariance for the Model for Prediction Across Scales–Atmosphere with the Joint Effort for data Assimilation Integration (JEDI-MPAS 2.0.0-beta)

Byoung-Joo Jung[1], Benjamin Ménétrier[2], Chris Snyder[1], Zhiquan Liu[1], Jonathan J. Guerrette[1], Junmei Ban[1], Ivette Hernández Baños[1], Yonggang G. Yu[1], and William C. Skamarock[1]

[1]National Center for Atmospheric Research, Boulder, Colorado 80301, USA
[2]Joint Center for Satellite Data Assimilation, Boulder, Colorado 80301, USA

**Correspondence:** Byoung-Joo Jung (bjung@ucar.edu)

**Abstract.** This paper describes the three-dimensional variational (3DVar) data assimilation (DA) system for the Model for Prediction Across Scales-Atmosphere with the Joint Effort for data Assimilation Integration (JEDI-MPAS). Its core element is a multivariate background error covariance implemented through multiple linear variable changes, including a wind variable change from stream function and velocity potential to zonal and meridional wind components, a vertical linear regression representing wind-mass balance, and multiplication by a diagonal matrix of error standard deviations. The univariate spatial correlations for the "unbalanced" variables utilize the Background error on an Unstructured Mesh Package (BUMP), which is one of generic components in the JEDI framework. The variable changes and univariate correlations are modeled directly on the native MPAS unstructured mesh. BUMP provides utilities to diagnose parameters of the covariance model, such as correlation lengths, from an ensemble of forecast differences, though some manual adjustment of the parameters is necessary because of mismatches between the univariate correlation function assumed by BUMP and the correlation structure in the sample of forecast differences. The resulting multivariate covariances, as revealed by single-observation tests, are qualitatively similar to those found in previous global 3DVar systems. Month-long cycling DA experiments using a global quasi-uniform 60 km mesh demonstrate that 3DVar, as expected, performs somewhat worse than a pure ensemble-based covariance, while a hybrid covariance that combines that used in 3DVar with the ensemble covariance, significantly outperforms both 3DVar and the pure ensemble covariance. Due to its simple workflow and minimal computational requirements, the JEDI-MPAS 3DVar can be useful for the research community.

## 1 Introduction

In the 1990s, three-dimensional variational (3DVar) data assimilation (DA) became the algorithm of choice in operational numerical weather prediction (Parrish and Derber, 1992; Andersson et al., 1998; Gauthier et al., 1999; Lorenc et al., 2000), owing to its numerous advantages relative to earlier optimal-interpolation assimilation schemes. 3DVar is no longer widely used operationally, both because its natural development path is to four-dimensional variational assimilation (Rabier et al., 2000; Rawlins





et al., 2007) and because of the rapid development of ensemble data assimilation in the last two decades, including ensemble-variational techniques that employ sample covariances from a forecast ensemble within the variational framework (Lorenc, 2003; Buehner, 2005). The central component of 3DVar systems, however, are so-called static covariance models (Bannister, 25 2008) that provide computationally tractable representations of complex spatial and multivariate covariances, and these remain in wide use to provide background covariances for 4DVar and as part of hybrid techniques that consider background covariance matrices that are the sum of a static covariance and an ensemble-based covariance (Hamill and Snyder, 2000).

This paper documents 3DVar and its associated static background covariance model for JEDI-MPAS, a data-assimilation system using software infrastructure from the Joint Effort for Data assimilation Integration (JEDI; Trémolet and Auligné, 30 2020) and the Model for Prediction Across Scales-Atmosphere (MPAS; Skamarock et al., 2012). Two companion papers are Liu et al. (2022), which gives an overview of JEDI-MPAS and initial results from a three-dimensional ensemble-variational (3DEnVar) scheme, and Guerrette et al. (2023), which documents an ensemble of data assimilations (EDA) for JEDI-MPAS.

Our motivation for implementing 3DVar is twofold. First, JEDI-MPAS is intended for use not only in our research, but also by the broader research community. The minimal computational cost of 3DVar and its simple workflow makes it well suited 35 where computing is a strong constraint and when introducing new users to the system. Experience with WRFDA (Barker et al., 2012), our existing community DA system, has shown that 3DVar is often preferred by users. Equally important, the static covariance model from 3DVar can be used in hybrid ensemble-variational assimilation schemes that are known to outperform 3DEnVar alone (Wang et al., 2008; Buehner et al., 2013; Clayton et al., 2013; Kuhl et al., 2013). We show the same result for JEDI-MPAS here (see also Guerrette et al., 2023).

The formulation of the static covariance model employed here has both familiar and novel elements. We generally follow Wu et al. (2002), including i) our choice of analysis variables, ii) the use of linear regression from a training data set to define the approximate mass-wind balances that implicitly determine the multivariate structure of the covariances (see also Derber and Bouttier, 1999), and iii) representing univariate correlations directly on the forecast model's grid (or mesh, in the case of MPAS). Products of vectors with the univariate spatial correlation matrices, however, are computed directly on a thinned subset 45 of the MPAS mesh and interpolated to the full-resolution mesh using the Background error on an Unstructured Mesh package (BUMP; Ménétrier, 2020). This study is the first evaluation of BUMP for use in atmospheric DA.

The outline of the paper is as follows. In the next section, we give an overview of JEDI-MPAS as configured for 3DVar. Section 3 describes the formulation of the static background covariances and their tuning using BUMP capabilities and a training data set of forecast differences. Section 4 presents single-observations tests that illustrate the structure of the implied 50 multivariate covariances. We summarize in section 5 results from cycling DA experiments with 3DVar and hybrid scheme, which provide an overall evaluation of the effectiveness of the JEDI-MPAS static covariances. Section 6 concludes and offers ideas for further refinements of the static covariances.





## 2 JEDI-MPAS 3DVar configuration

### 2.1 The forecast model

MPAS is described in detail in Skamarock et al. (2012) or see Liu et al. (2022) for a more concise summary. Briefly, MPAS integrates the nonhydrostatic equations of motion cast in a heigh-based, terrain-following vertical coordinate and using dry density and a modified moist potential temperature as thermodynamic variables. The equations are discretized on an unstructured mesh with the normal component of horizontal velocity defined on the edges of mesh cells and other prognostic variables defined at the cell centers. MPAS supports global and regional meshes, as well as meshes with quasi-uniform or variable resolution.

In all experiments presented here, MPAS is configured with a global quasi-uniform mesh of 60 km resolution and 55 vertical levels up to a model top of 30 km. The physical parameterizations are those of the "mesoscale reference" suite, as listed in table 2 of Liu et al. (2022).

### 2.2 The DA system

JEDI-MPAS implements various abstract classes for MPAS within the JEDI framework (Trémolet, 2020). Those abstract
classes reside in the Object Oriented Prediction Systems (OOPS) and comprise all the building blocks and operations on them necessary for data assimilation algorithms. JEDI also contains generic (model independent) implementations of some building blocks, including observation operators and quality control (the United Forward Operator, UFO), observation storage and access (the Interface for Observation Data Access, IODA), the background-error covariance matrix (the System-Agnostic Background Error Representation, SABER).

The variational application of OOPS minimizes a cost function given in (3) of Liu et al. (2022). Denoting by $\boldsymbol{x}$ the concatenation the analysis variables across all model levels and mesh locations, the cost function measures the simultaneous fit of $\boldsymbol{x}$ to a background state $\boldsymbol{x}_b$, which is our best estimate of $\boldsymbol{x}$ before considering the observations, and to the observations $\boldsymbol{y}$ (also concatenated into a single vector) via an observation operator $h(\boldsymbol{x})$ that maps a given state to the observation variables.

The minimization proceeds iteratively by linearizing the observation operator in the neighborhood of the latest iterate $\boldsymbol{x}_g$
and computing the next iterate as the minimizer of the resulting quadratic cost function,

$$J(\delta\boldsymbol{x}) = \frac{1}{2}(\delta\boldsymbol{x} - \delta\boldsymbol{x}_g)^T \mathbf{B}^{-1}(\delta\boldsymbol{x} - \delta\boldsymbol{x}_g) + \frac{1}{2}(\mathbf{H}\delta\boldsymbol{x} - \boldsymbol{d})^T \mathbf{R}^{-1}(\mathbf{H}\delta\boldsymbol{x} - \boldsymbol{d}), \tag{1}$$

where $\delta\boldsymbol{x} = \boldsymbol{x} - \boldsymbol{x}_g$ is the increment relative to $\boldsymbol{x}_g$, $\delta\boldsymbol{x}_g = \boldsymbol{x}_b - \boldsymbol{x}_g$. $\boldsymbol{d} = \boldsymbol{y} - h(\boldsymbol{x}_g)$ is the observation departure from $\boldsymbol{x}_g$, $\mathbf{H}$ is the linearization of $h$ near $\boldsymbol{x}_g$, and $\mathbf{B}$ and $\mathbf{R}$ are the background and observational error covariance matrices, respectively. This incremental formulation is central to the architecture of OOPS and distinguishes increments, which can be operated on by $\mathbf{B}$
and $\mathbf{H}$, from the full state, which is an argument to $h$. In what follows, increments will be indicated by variables preceded with $\delta$.

All the minimization schemes for (1) and preconditioners available in OOPS involve only the application of $\mathbf{B}$ to increments, rather than its square root or inverse. For the single observation test and cycling experiments shown later, we employ the $\mathbf{B}$-





preconditioned incremental variational application of OOPS and the Derber-Rosati Inexact Preconditioned Conjugate Gradient algorithm (Golub and Ye, 1999; Derber and Rosati, 1989).

## 2.3 Analysis variables and variable change

The analysis variables are the horizontal velocity ($\boldsymbol{v}$), temperature ($T$), specific humidity ($q$), and surface pressure $p_s$ at the MPAS cell centers, as described in Liu et al. (2022). Transformations to other variables are necessary for some observation operators and for initial conditions for MPAS forecasts. Those transformations also follow Liu et al. (2022) but with one significant improvement.

In Liu et al. (2022), pressure $p$ is computed hydrostatically from the full $T$, $p_s$, and $q$, the dry density $\rho_d$ is given by equation of state knowing $T$, $q$, and $p$, and the dry potential temperature $\theta_d$ is computed from its definition given $T$ and $p$. This approach meant that $p$, $\rho_d$, and $\theta_d$ would be changed relative to the background forecast—owing to nonhydrostatic effects in the model and to discretization errors in the vertical integral used to compute $p$ from the hydrostatic relation—even if no observations were assimilated and increments for the analysis variables were zero.

Here, we instead compute increments for $p$, $\rho_d$, and $\theta_d$ from the *increments $\delta T$, $\delta p_s$, and $\delta q$*, by linearizing the corresponding calculations of Liu et al. (2022, steps 3 and 4 of their section 3.3). The new, linearized update for $p$ and $\rho_d$ has reduced temperature bias in the stratosphere, especially near the model top, in JEDI-MPAS cycling experiments (not shown) and will be part of the next JEDI-MPAS release.

The state ($\boldsymbol{x}$) and increment ($\delta \boldsymbol{x}$) objects in JEDI-MPAS are basically inherited from MPAS's "pool type" data structure. Thus, it is natural to choose the existing mesh decomposition and communication utilities of MPAS to handle the parallelism of the state and increment of JEDI-MPAS. The state and increment objects in JEDI-MPAS only contain their values on own grid point without a halo region.

## 3 Multivariate background error covariance

In this section, we will present how the multivariate static background error covariance is designed for JEDI-MPAS. With a series of linear variable changes, the JEDI-MPAS analysis variables are transferred into a set of variables that are independent each other. Then, we will describe how the $\mathbf{B}$ statistics (or parameters) can be trained from samples. The characteristics of diagnosed $\mathbf{B}$ statistics at MPAS 60 km uniform mesh will be discussed. Lastly, we will discuss what modification is made to the diagnosed $\mathbf{B}$ statistics to resolve the discrepancy between the assumption and actual sample dataset.

## 3.1 Multivariate background error covariance design

The basic design of the JEDI-MPAS multivariate background error covariance follows that of the Gridpoint Statistical Interpolation (GSI; Wu et al. (2002)) system, except in our use of BUMP, rather than recursive filters, to model the univariate spatial correlations. The multivariate covariances are implemented as two linear variable changes, $\mathbf{K}_1$ and $\mathbf{K}_2$, applied to a





block-diagonal covariance matrix

$$\mathbf{B} = \mathbf{K}_1 \mathbf{K}_2 \mathbf{\Sigma} \mathbf{C} \mathbf{\Sigma} \mathbf{K}_2^T \mathbf{K}_1^T, \tag{2}$$

where the block diagonal covariance matrix has been written as the product of a block-diagonal correlation matrix $\mathbf{C}$ and a diagonal matrix $\mathbf{\Sigma}$ of standard deviations.

The linear variable changes $\mathbf{K}_1$ and $\mathbf{K}_2$ can be expressed in the following matrix forms:

$$\mathbf{K}_1 : \begin{bmatrix} \delta\boldsymbol{v} \\ \delta T \\ \delta q \\ \delta p_s \end{bmatrix} = \begin{bmatrix} \boldsymbol{k} \times \nabla & -\nabla & 0 & 0 & 0 \\ 0 & 0 & \mathbf{I} & 0 & 0 \\ 0 & 0 & 0 & \mathbf{I} & 0 \\ 0 & 0 & 0 & 0 & \mathbf{I} \end{bmatrix} \begin{bmatrix} \delta\psi \\ \delta\chi \\ \delta T \\ \delta q \\ \delta p_s \end{bmatrix} \tag{3}$$

and

$$\mathbf{K}_2 : \begin{bmatrix} \delta\psi \\ \delta\chi \\ \delta T \\ \delta q \\ \delta p_s \end{bmatrix} = \begin{bmatrix} \mathbf{I} & 0 & 0 & 0 & 0 \\ \mathbf{L} & \mathbf{I} & 0 & 0 & 0 \\ \mathbf{M} & 0 & \mathbf{I} & 0 & 0 \\ 0 & 0 & 0 & \mathbf{I} & 0 \\ \mathbf{N} & 0 & 0 & 0 & \mathbf{I} \end{bmatrix} \begin{bmatrix} \delta\psi \\ \delta\chi_u \\ \delta T_u \\ \delta q \\ \delta p_{s,u} \end{bmatrix} \tag{4}$$

.

Here, $\mathbf{K}_1$ computes increments for $\boldsymbol{v}$ from spatial derivatives (indicated schematically by the gradient terms in the upper-left block) of increments of stream function $\psi$ and velocity potential $\chi$. $\mathbf{K}_1$ and the corresponding adjoint operator, $\mathbf{K}_1^T$, are model-dependent JEDI components that operate on the MPAS native mesh and are implemented in "*Control2Analysis*", a linear variable change class. Details of the calculation of $\boldsymbol{v}$ from $\psi$ and $\chi$ are given in the appendix.

$\mathbf{K}_2$ is a linear variable change that computes $\delta\chi$, $\delta T$, and $\delta p_s$ from $\delta\psi$ and the residual or "unbalanced" increments $\delta\chi_u$, $\delta T_u$, and $\delta p_{s,u}$ (for velocity potential, temperature, and surface pressure, respectively), so called because they are by assumption independent of $\delta\psi$. The relation of $\delta\chi$, $\delta T$, and $\delta p_s$ to $\delta\psi$ is based on linear regression from a training data set, following Derber and Bouttier (1999). As in Derber and Bouttier (1999), we choose to use $\delta\psi$ at a given mesh cell and on the full set of vertical levels as predictors for $\delta T$ and $\delta p_s$ at the same mesh cell and on any specific level, which makes $\mathbf{M}$ and $\mathbf{N}$ block diagonal with blocks corresponding to mesh cells and full matrices in each block. We retain $\delta\psi$ only on the same level as a predictor for $\delta\chi$, which makes $\mathbf{L}$ a diagonal matrix.

Lastly, $\mathbf{\Sigma}\mathbf{C}\mathbf{\Sigma}^T$ is the covariance matrix for $\delta\psi$, $\delta q$, and the unbalanced increments $\delta\chi_u$, $\delta T_u$, and $\delta p_{s,u}$. We assume these variables are mutually independent, so that $\mathbf{C}$ is block diagonal with blocks that give the univariate spatial (horizontal and vertical) correlation for each variable. The matrix $\mathbf{\Sigma}$ is a diagonal matrix with elements that specify the standard deviation for $\delta\psi$, $\delta\chi_u$, $\delta T_u$, $dq$, and $\delta p_{s,u}$.





The operations $\mathbf{K}_2$, $\mathbf{K}_2^T$, $\mathbf{\Sigma}$, $\mathbf{\Sigma}^T$, and $\mathbf{C}$ use the Background error on Unstructured Mesh Package (BUMP) in the System-Agnostic Background-Error Representation (SABER) repository, which is a generic component within JEDI, through the
MPAS model interfaces. The BUMP Vertical BALance (VBAL) driver is used for $\mathbf{K}_2$ and $\mathbf{K}_2^T$, and the BUMP VARiance (VAR) driver is used for $\mathbf{\Sigma}$ and $\mathbf{\Sigma}^T$. The spatial correlation matrix is pre-computed from the given correlation lengths with BUMP Normalized Interpolated Convolution from on Adaptive Subgrid (NICAS; Ménétrier, 2020) driver. NICAS can efficiently represent the large correlation matrix on the full grid with an equivalent correlation matrix on a subgrid (which is subsampled from the full grid) and the linear interpolation between full and subgrid.

## 145  3.2  Training the covariance model

The designed multivariate background error covariance has several parameters to be determined. These parameters are diagnosed from 366 samples of National Centers for Environmental Prediction (NCEP) Global Forecast System (GFS) 48-h and 24-h forecast differences, valid at the same time of day and spanning the months of March, April and May 2018.

Since $\mathbf{\Sigma}\mathbf{C}\mathbf{\Sigma}^T$ depends on the statistics of $\delta\psi$ and $\delta\chi$, we first need to calculate perturbations of $\psi$ and $\chi$ from $\delta\boldsymbol{v}$ of each
forecast difference. This is essentially the inverse operation to $\mathbf{K}_1$ and can be expressed as solving a Poisson equation with vorticity or divergence as a source term. Because solving a Poisson equation efficiently on the unstructured grid is not an easy task, we have adopted a spherical-harmonics-based function from the NCAR Command Language (NCL, 2019) that operates on an intermediate latitude-longitude grid. We begin by interpolating $\delta\boldsymbol{v}$ fields to the intermediate grid, and then calculate $\delta\psi$ and $\delta\chi$ through the "*uv2sfvpf*" function of NCL and interpolate back to the MPAS mesh. Note that because the definition of
$\delta\chi$ (shown in Eq. 3) is opposite in sign with the definition of NCL function, multiplying $(-1)$ to $\delta\chi$ (from NCL) is required.

For $\mathbf{K}_2$, the BUMP VBAL driver calculates the cross-variable linear regression coefficients, which are denoted as $\mathbf{L}$, $\mathbf{M}$, and $\mathbf{N}$ in Eq. 4. The vertical autocovariance matrix of $\delta\psi$ and the vertical cross-covariance matrices between $\delta\psi$ and each of $\delta\chi$, $\delta T$, and $\delta p_s$ are computed and averaged over latitude bands of $\pm 10$ degrees. The desired matrices of regression coefficients are obtained in the standard way by right multiplying the cross-variable covariance by the inverse autocovariance of the predictor
variable ($\delta\psi$ in our design). The vertical autocovariance matrices are usually poorly conditioned and thus direct computation of their inverses will yield noisy results in the presence of sampling error. To overcome this, we apply a pseudo-inverse, which only includes some dominant eigenmodes to calculate the inverse matrix. We have chosen the leading 20 modes (among total 55 modes) for the pseudo-inverse of the $\delta\psi$ autocovariance matrix.

For $\mathbf{\Sigma}$, the BUMP VAR driver calculates variances for $\delta\psi$, $\delta\chi_u$, $\delta T_u$, $\delta q$, and $\delta p_{s,u}$.

The correlation matrix, $\mathbf{C}$, consists of blocks that specify the univariate spatial correlation for $\delta\psi$, $\delta\chi_u$, $\delta T_u$, $\delta q$, and $\delta p_{s,u}$. The BUMP Hybrid DIAGnostic (HDIAG) driver diagnoses the horizontal and vertical correlation lengths used in modeling $\mathbf{C}$ parameters. HDIAG calculates the sample correlations from different separation distances, then fits the sample correlation curves to the fifth-order piecewise function of Gaspari and Cohn (1999), which resembles the Gaussian function, assumed in NICAS to get the correlation length parameters. The vertical distance in these calculations is measured by the difference in
horizontally averaged height between two vertical levels. In BUMP, the correlation length parameter represents the compact support radius, beyond which the correlation becomes zero.





### 3.3 Diagnosed statistics

The regression coefficients that appear in the definition (4) of $\mathbf{K}_2$, which are computed by BUMP VBAL, are shown in Fig. 1 for a location near $34.8°$ N latitude. Considering first Fig. 1a, the $\delta T$–$\delta \psi$ coefficients are largest at small separations. Their
structure is dipolar, with $\delta T$ at a given level positively related to $\delta \psi$ at nearby but higher levels, and negatively related to $\delta \psi$ at nearby but lower levels. The $\delta T$–$\delta \psi$ coefficients are generally consistent with approximate geostrophic and hydrostatic balance, which together relate $\psi$ to the mass field and the vertical derivative of the mass field to buoyancy. The coefficient structure is different for model levels lower than 10, perhaps due to the effects of the boundary layer and terrain. Figure 1b shows $\delta \chi$–$\delta \psi$ coefficients, which relate $\delta \chi$ at a given level to $\delta \psi$ at the same level. The balanced part of $\delta \chi$ depends positively on $\delta \psi$ near
the surface, consistent with Ekman balance, under which vertical vorticity near the surface drives horizontal convergence in the boundary layer. Finally and not unexpectedly, $\delta p_s$ has a positive dependence on $\delta \psi$ in the lower troposphere (Fig. 1c).

     Figure 2 shows the variance that can be predicted knowing $\delta \psi$ normalized by the total variance, for $\delta T$, $\delta \chi$, and $\delta p_s$. There are substantial variations with latitude and height. For $\delta T$, the $\delta T$–$\delta \psi$ regression can explain up to 70 % of the total variance in mid- and high-latitude regions in the troposphere. For $\delta \chi$, the regression explains up to 35 % of the total variance in the
mid-latitude near the surface (below model level 10), while for $\delta p_s$, the regression explains substantial variance everywhere except the tropics. These balanced ratios are similar to those found in Wu et al. (2002) (their Fig.1) and Barker et al. (2012) (their Fig. 5), and their geographic variations confirm that the regressions primarily reflect dynamical balances characteristic of mid- and high latitudes.

     The other quantities that must be estimated from the training data set are the standard deviations that form the diagonal of $\boldsymbol{\Sigma}$
and the fields of horizontal and vertical correlation scales for each of the variables $\delta \psi$, $\delta \chi_u$, $\delta T_u$, $\delta q$ and $\delta p_{s,u}$, which together specify the correlation matrix $\mathbf{C}$. Figure 3 shows the vertical profiles of horizontally averaged standard deviations for each variable. For $\delta \psi$ and $\delta \chi_u$, the standard deviation increases upward from the surface to a peak near the tropopause. The standard deviation of $\delta T_u$, in contrast, generally decreases upward from a peak at the surface. For $\delta q$, the profile of standard deviations has a shape similar to that for $q$ itself, peaked in the lower troposphere and decreasing steadily with height above.

Figures 4 and 5 show the vertical profiles of horizontally averaged horizontal and vertical correlation lengths, respectively. The horizontal correlation lengths generally increase with height in the stratosphere and nearly constant in the troposphere, though $\delta \chi_u$ has substantial variations in horizontal length scale throughout the profile. The horizontal correlation lengths for $\delta \psi$ and $\delta \chi_u$ are larger than those for $\delta T_u$ and $\delta q$, while the horizontal correlation length for $\delta p_{s,u}$ is roughly 3700 km, much larger than the horizontal correlation lengths for $\delta T_u$ and $\delta q$ near the surface. The vertical correlation lengths have minima near
the surface for all variables and then increase with height toward a peak near the model top. The vertical correlation lengths for $\delta \psi$ and $\delta \chi_u$ are again larger than those for $\delta T_u$ and $\delta q$.





### 3.4 Additional tuning

The parameters shown in the previous section are the raw statistics from the BUMP training applications. We have applied two additional modifications to these raw statistics. Without these modifications, the resulting static $\mathbf{B}$ performs poorly in 3DVar and is unable to improve on 3DEnVar in hybrid applications (not shown).

First, the background error standard deviation for all variables ($\mathbf{\Sigma}$) is scaled by a factor of 1/3. While the cycling interval shown later in this study is 6 hours, which is typical, the forecast differences used in the training reflect forecast-error evolution over 24 hours. Thus, it is reasonable to scale the diagnosed background error standard deviation to match the error growth for a 6 hour interval. Here, we choose the single scaling factor of 1/3 for all variables, based on a limited set of sensitivity tests of cycling experiments with different scaling factors.

We also reduce the diagnosed horizontal correlation lengths for $\delta\psi$ and $\delta\chi_u$ by half. Figure 6 shows the raw horizontal correlation function for $\delta\psi$ on model level 15 together with the best-fit correlation function, which is diagnosed by BUMP by adjusting the length scale of the fifth-order, compactly supported function from Gaspari and Cohn (1999). There is a clear discrepancy between the sample correlation function and that assumed in BUMP—the raw correlation decreases more rapidly for small separations and has larger correlations at large separations. Since the implied velocity variance depends on the second derivative at the origin of the $\delta\psi$ (and $\delta\chi$) correlation (Lorenc, 1981; Daley, 1985), the diagnosed covariances greatly underestimate the velocity variance relative to the statistics of the original training data. Reducing the horizontal correlation length for $\delta\psi$ and $\delta\chi_u$ increases the velocity variances, though at the expense of further underestimating the correlations at larger separations.

Ideally, the mismatch between the assumed correlation structure and that of the training data would be addressed by a more flexible correlation model in BUMP. A capability is now available and we hope to report on its use in the future. In other systems, the necessary flexibility has been achieved by using sums of recursive filters with different length scales (Wu et al., 2002; Kleist et al., 2009) or modeling the correlations in spectral space (Parrish and Derber, 1992; Lorenc et al., 2000).

## 4 Single Observation Tests

To explore the structure of diagnosed and tuned multivariate B, two sets of single observation test were performed; one for assimilating a single zonal wind observation with $1 \text{ ms}^{-1}$ innovation and $1 \text{ ms}^{-1}$ observation-error standard deviation, and the other for assimilating a single temperature observation with 1 K innovation and 1 K observation-error standard deviation. Both single observations are placed at $(38.68° \text{ W}, 40.4113° \text{ N})$ and at roughly $800 \text{ hPa}$, a location that corresponds to one of the MPAS cell centers and vertical level 15.

Figure 7 shows the analysis increments from the single temperature observation, for the zonal and meridional components of $\boldsymbol{v}$ (left and center columns, respectively) and for $T$ (right column), on the three different vertical levels (10, 15, and 20, shown in the bottom, middle and top rows) . The $T$ increments have a horizontally isotropic structure with maximum values at level 15, where the observation is located. The wind increments are, to a first approximation, linked to the $T$ increment through the thermal-wind relation: cyclonic circulation is introduced on level 10, below the maximum temperature increment, an anti-





cyclonic circulation appears above, on level 20. This reflects the approximate geostrophic and hydrostatic relations captured by $\mathbf{K}_2$, and is consistent with Parrish and Derber (1992) (their Fig. 2), which uses the linear balance equation between mass and momentum variables.

Similarly, Fig. 8 shows the analysis increments from the single zonal-wind observation. The positive zonal-wind increment is maximized at the observation location on the 15th model level. To the north and south of the observation location, negative

zonal-wind increments are introduced, which—together with the increments of meridional winds—represent a cyclonic circulation to the north of the observation and an anti-cyclonic circulation to the south. Temperature increments are negative below the cyclonic circulation (i.e., on level 10) and positive above (on level 20), and vice versa for the anti-cyclonic circulation. The structure of the increments again reflects thermal-wind balance, and in this case is consistent with Wu et al. (2002) (their Fig. 9) or Kleist et al. (2009) (their Fig. 3).

Figure 9 shows the surface pressure increments from two single observation tests. When the single temperature observation is assimilated, cyclonic circulation is introduced in the lower troposphere. The negative surface pressure increment is approximately geostrophically related to this cyclonic circulation. When the single zonal wind observation is assimilated, the zonal wind increments extend throughout the troposphere, including to the surface. The dipole of positive and negative surface pressure increments, south and north respectively of the observation location, are geostrophically related to the increment of the

surface wind.

## 5 Cycling Experiments

### 5.1 Experimental design and assimilated observations

For further evaluation of the multivariate static $\mathbf{B}$ for JEDI-MPAS, three sets of one-month (15 April–14 May 2018) cycling experiments were performed on NCAR's High Performance Computing system, Cheyenne. As a reference experiment, the

"3DEnVar" experiment was performed using the pure ensemble covariances, as in Liu et al. (2022). At each cycle, a 20-member ensemble of 6-hour MPAS forecasts was performed using initial conditions from the Global Ensemble Forecast System (GEFS; Zhou et al., 2017)). Covariance localization was applied to the ensemble covariances via BUMP's generic localization scheme, using globally constant localization scales of 1200 km horizontally and 6 km vertically. To demonstrate the static covariances, the "3DVar" experiment was performed with the static $\mathbf{B}$ formulated and tuned as described in the preceding sections. Lastly,

the "Hybrid-3DEnVar" experiment was performed using a hybrid covariance given by a weighted sum of static and ensemble $\mathbf{B}$ (Hamill and Snyder, 2000). Here, we choose a weight of 0.5 for each component, similar to Wang et al. (2013), Clayton et al. (2013), and Kuhl et al. (2013). This final experiment evaluates the effectiveness of our static $\mathbf{B}$ for hybrid applications. In all three experiments, the same global MPAS quasi-uniform 60 km mesh is used both for analysis and background fields and for analysis increment. For the minimization, two outer loops are used, with 60 inner iterations for each outer loop.

The observation files are converted from GSI's ncdiag files, which contains the observation location, observation value, observation error, GSI's quality control, and satellite bias correction information. The observation quality control basically follows the GSI's quality control (called "PREQC"), and the background innovation check is added, which filters out the



observation when the absolute value of observation departure is larger than three times of given observation error. In all three experiments, the surface pressure, radiosondes (wind, temperature or virtual temperature, and specific humidity), aircraft (wind, temperature, specific humidity), atmospheric motion vectors, Global Navigation Satellite System Radio Occultation (GNSS RO) refractivity, and clear-sky Advanced Microwave Sounding Unit-A (AMSU-A) radiances from NOAA-15/18/19, METOP-A/B, and Aqua satellites are assimilated. The AMSU-A radiances are bias-corrected from GSI's information and pre-thinned with 145 km mesh.

## 5.2 Results

Figure 10 shows the time series of background root mean square errors (RMSE)s for surface pressure during the cycling period. The RMSEs are calculated with respect to GFS analysis at the valid time as a reference. For surface pressure, 3DVar gives somewhat smaller RMSEs over the 3DEnVar experiment, except for the southern extratropical region. Hybrid-3DEnVar gives smaller RMSEs over 3DVar. Figure 11 shows the vertical profiles of relative RMSE changes for background fields during the cycling period, with the RMSE of 3DEnVar as reference. The confidence intervals with 95 % significant level are also shown as error bars, from bootstrap resampling method with resampling size of 10000. Compared to 3DEnVar, the 3DVar experiment shows some degradation in the troposphere and some improvement in the stratosphere in general. The Hybrid-3DVar shows overall improvement over both 3DEnVar and 3DVar experiments, and throughout levels in both the troposphere and stratosphere.

Figure 12 shows the observation space verification for radiosonde. The relative change of root mean square (RMS) first-guess departure (OMB, observation minus background) are mostly consistent with model space verification in Fig. 11. Compared to the RMSs of 3DEnVar, the RMSs of Hybrid-3DEnVar are significantly improved, except for temperature observation. The RMSEs of 3DVar are degraded by ∼ 5 %. In the observation space verification for AMSU-A radiance observations (not shown), which assimilates the channels sensitive to the atmospheric temperature profile, the Hybrid-3DEnVar shows neutral to slightly improved impact in the RMSEs for channels 5 and 6. For channels 7, 8, and 9, both 3DVar and Hybrid-3DEnVar show significant improvement over 3DEnVar. Larger improvement is shown over both high latitudes. This is consistent with the large temperature RMSE reduction in the model-space verification (Fig. 11a).

Additional 10-day extended forecasts were conducted at each 00:00 UTC initialization time to evaluate the impact of analysis on the longer forecast lengths. The changes in RMSE for 3DVar and Hybrid-3DEnVar relative to 3DEnVar are shown in Fig. 13. At short forecast lead times, the relative RMSEs look similar to the relative RMSEs for the 6-h background forecasts shown in Figs. 10 and 11. The benefit of hybrid background error covariance can be found up-to 5 day lead time for surface pressure, temperature, and zonal and meridional wind fields. The benefit of hybrid covariance is only kept for ∼ 2 day lead time for humidity fields.



## 6   Conclusions

This study has described the multivariate static background error covariances for JEDI-MPAS 3DVar. Similar to Liu et al.
(2022), JEDI-MPAS 3DVar utilizes generic JEDI components, through interfaces that are specific to MPAS.

The formulation of the JEDI-MPAS static $\mathbf{B}$ generally follows Wu et al. (2002), but with the novel use of BUMP for multiple elements of the covariance model. Two linear variable transforms represent the multivariate relationship. One transform is a variable change from streamfunction $\psi$ and velocity potential $\chi$ to $\boldsymbol{v}$, which directly operates on the MPAS native mesh. The other transform, which uses the BUMP driver VBAL from JEDI's SABER repository, is based on linear regression over vertical
columns of increments in other variables against increments in $\psi$. The full multivariate covariances are then given by these linear transforms (and their adjoints) applied to a univariate covariance. The univariate correlation matrix employs BUMP NICAS, which efficiently computes the three-dimensional convolution of an input vector with a specified spatial correlation function on an optimally subsampled mesh and then interpolates back to the full MPAS mesh.

For the experiments presented here, we estimated various parameters in the covariance model from a training dataset of
366 differences between 48-h and 24-h forecasts on the 60-km MPAS mesh. In general, the regression coefficients capture the linear, approximately geostrophic balance between mass and momentum variables that holds outside the tropics. While the error standard deviations for $\psi$ and $\chi_u$ get larger at higher vertical levels, up to a peak near the tropopause, the error standard deviations for $T_u$ and $q$ are larger in the mid- to lower troposphere. The horizontal and vertical correlation lengths for errors in $\psi$ and $\chi_u$ are, in general, larger than those for errors in $T_u$ and $q$. We also made two modifications to the parameters
objectively estimated by BUMP. The error standard deviation is scaled by a factor of 1/3 to match the 24 hour time differences (i.e., 48 h forecast and 24 forecast pairs) in the training samples to the typical 6 hour DA cycling interval. In addition, the horizontal correlation lengths for increments of $\psi$ and $\chi_u$ are reduced by half to compensate for the discrepancy between the raw correlation structure from the training data set and the correlation function assumed in BUMP, which has much less curvature at small separations.

We evaluated the JEDI-MPAS static $\mathbf{B}$ in cycling data-assimilation experiments extending over a month and assimilating observations from a significant fraction of the global observing network, including GNSS RO, AMSU-A and conventional observations. 3DVar using this static $\mathbf{B}$ generally performs close to, but worse than, EnVar using purely ensemble covariances as in Liu et al. (2022), while using a hybrid background covariance that is a weighted sum of the static $\mathbf{B}$ and ensemble covariances improves significantly on both 3DVar and EnVar. Neither of these results is novel, as numerous studies have shown
the advantage of EnVar over 3DVar and of the hybrid algorithm over EnVar, but they do demonstrate clearly the effectiveness of the static $\mathbf{B}$.

The static background covariances presented here are an initial implementation, with plenty of room for further refinements. Two extensions that are already under way are training the covariance model based on an ensemble from JEDI-MPAS, such as those provided by the EDA of Guerrette et al. (2023), and including hydrometeor increments, which will be especially
important for all-sky assimilation of radiances. There are also BUMP capabilities that we have yet to exercise, including more



general correlation functions that should remove the need for manual retuning of correlation lengths diagnosed by BUMP, and joint estimation of hybridization and localization coefficients (Ménétrier and Auligné, 2015).

*Code availability.* JEDI-MPAS 2.0.0-beta is publicly released on GitHub, accessible in the release/2.0.0-beta branch of mpas-bundle (https://github.com/JCSDA/mpas-bundle/tree/release/2.0.0-beta). It is also available from Zenodo at https://doi.org/10.5281/zenodo.7630054 (Joint
Center For Satellite Data Assimilation and National Center For Atmospheric Research, 2022). Global Forecast System analysis data are downloaded from NCAR Research Data Archive https://rda.ucar.edu/datasets/ds084.1/ (last access: 1 June 2023; National Centers For Environmental Prediction/National Weather Service/NOAA/U.S. Department Of Commerce, 2015). Global Ensemble Forecast System ensemble analysis data are downloaded from https://www.ncei.noaa.gov/products/weather-climate-models/global-ensemble-forecast (last access: 1 June 2023). Conventional and satellite observations assimilated are downloaded from https://rda.ucar.edu/datasets/ds337.0/ (last
access: 1 June 2023; National Centers For Environmental Prediction/National Weather Service/NOAA/U.S. Department Of Commerce, 2008) and https://rda.ucar.edu/datasets/ds735.0/ (last access: 1 June 2023; National Centers For Environmental Prediction/National Weather Service/NOAA/U.S. Department Of Commerce, 2009).

## Appendix A: Diagnosing velocity from stream function and velocity potential in MPAS

To compute horizontal velocity $\boldsymbol{v}$ from $\psi$ and $\chi$ on the MPAS mesh, we rely on the fact that the irrotational component of
velocity is given by (minus) the gradient of $\chi$, while the nondivergent component of velocity is given by cross product of the vertical unit vector and the gradient of $\psi$. The edge-normal component of velocity $u$ on the MPAS mesh is oriented parallel to the segment connecting the centers of adjacent cells and, naturally, normal to the edge itself. The natural finite difference relation is then

$$u = -\delta_c \chi - \delta_v \psi, \tag{A1}$$

where, in an abuse of our previous notation, $\delta_c$ is the difference operator between the centers of the mesh cells adjoining the edge and $\delta_v$ is the difference operator between the cell vertices at either end of the edge.

Implementing (A1) is straightforward given the MPAS mesh conventions and the mesh information in MPAS initialization files (MPA, 2015). The difference operators are defined as

$$\begin{aligned} \delta_c \chi &= \frac{\chi_{c,2} - \chi_{c,1}}{\Delta c}, \\ \delta_v \psi &= \frac{\psi_{v,2} - \psi_{v,1}}{\Delta v}, \end{aligned} \tag{A2}$$

where $\Delta c$ is the distance between cell centers sharing the edge, and $\Delta v$ is the distance between vertices on the edge. The ordering of the cells and vertices are such that this formula will give the correct signs for the velocities on the MPAS mesh. For a given edge in the files, the lengths $\Delta c$ and $\Delta v$ are in the variables $dcEdge(edge)$ and $dvEdge(edge)$, respectively. The cells sharing an edge are $cellsOnEdge(2, edge)$ and $cellsOnEdge(1, edge)$, and the vertices are $verticesOnEdge(2, edge)$ and $verticesOnEdge(1, edge)$.





Although the equations A1 and A2 are using the full variables (i.e., $\psi$ and $\chi$), they are also applicable to the incremental variables (i.e., $\delta\psi$ and $\delta\chi$) because of their linear form. The computation begins with $\delta\psi$ and $\delta\chi$ at the cell centers. Values of $\delta\psi$ at the cell vertices are computed by interpolating $\delta\psi$ from the centers of the three cells containing each vertex, before applying (A1). MPAS provides a utility that employs radial basis functions to reconstruct the vector wind at a cell center from $\delta u$ on the edges of the cell (following Bonaventura et al., 2011), which provides the final $\delta\boldsymbol{v}$. Because these three steps involve

different subset of the MPAS mesh (i.e., cells, vertices, and edges), a halo exchange routine from MPAS is required and called between each step. We have also implemented the adjoint operator of the halo exchange, which is needed when applying $\mathbf{K}_1^T$.

*Author contributions.* The first author designed, conducted, and analyzed all experiments, and wrote the manuscript. Benjamin Ménétrier gave a feedback to configure the BUMP drivers and developed the necessary component in BUMP. Chris Snyder and Zhiquan Liu contributed the design of multivariate covariance formulation and analysis of experimental results. William C. Skamarock provided the discretized equa-

tion for wind transform shown in appendix. All co-authors contributed to the development of the JEDI-MPAS source code and experimental workflow, preparation of externally sourced data, design of experiments, and preparation of the manuscript.

*Competing interests.* The authors have no competing interests.

*Acknowledgements.* The National Center for Atmospheric Research is sponsored by the National Science Foundation of the United States. This research has been supported by the United States Air Force (grant no. NA21OAR4310383). We would like to acknowledge high

performance computing support from Cheyenne (doi:10.5065/D6RX99HX) provided by NCAR's Computational and Information Systems Laboratory, sponsored by the National Science Foundation. Michael Duda in the Mesoscale and Microscale Meteorology (MMM) laboratory provided guidance on modifying MPAS-A for JEDI-MPAS applications.



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





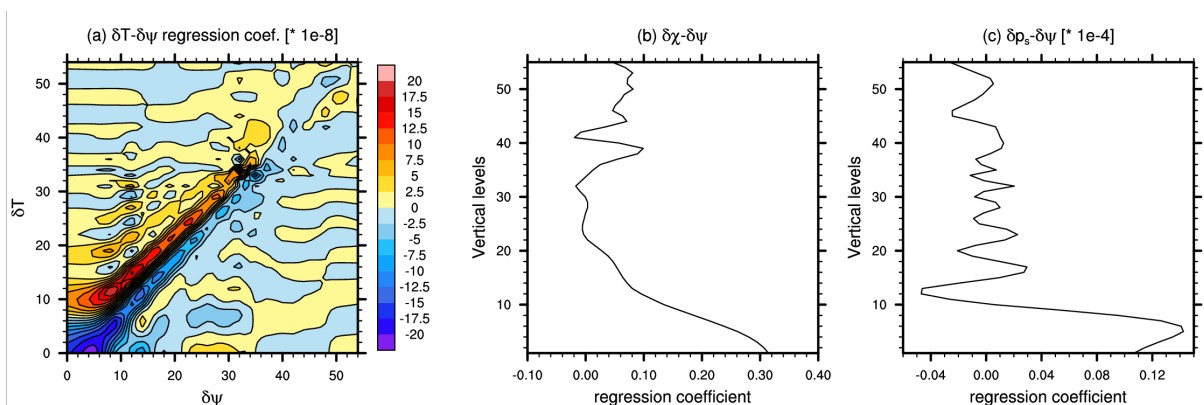

**Figure 1.** Regression coefficients near $34.8°$ N between (a) $\delta T$ and $\delta \psi$ $[K(m^2 s^{-1})^{-1}]$, (b) $\delta \chi$ and $\delta \psi$ $[(m^2 s^{-1})(m^2 s^{-1})^{-1}]$, and (c) $\delta p_s$ and $\delta \psi$ $[Pa(m^2 s^{-1})^{-1}]$. These are the nonzero elements at this mesh cell of the submatrices $\mathbf{L}$, $\mathbf{M}$ and $\mathbf{N}$, repsectively, of $\mathbf{K}_2$ [see Eq. (4)].

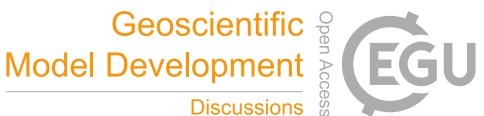

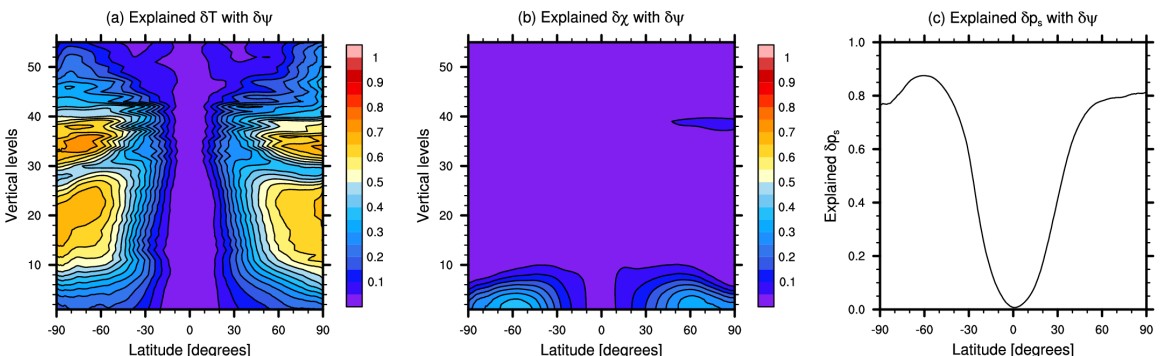

**Figure 2.** Ratio of balanced variance (i.e. that predicted by $\delta\psi$) to total variance for (a) $\delta T$, (b) $\delta\chi$, and (c) $\delta p_s$.



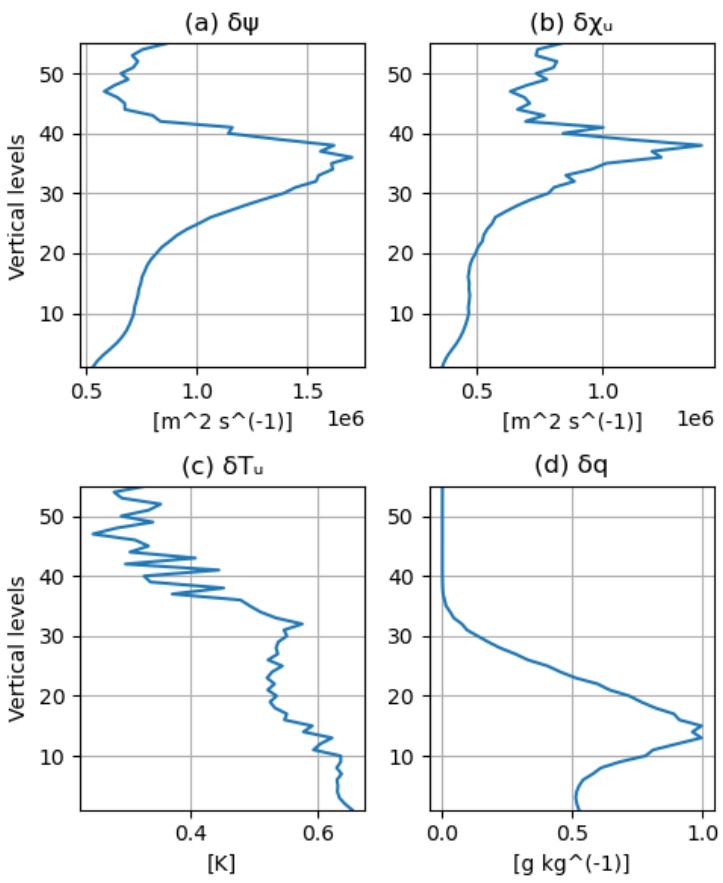

**Figure 3.** Horizontally averaged standard deviations for (a) $\delta\psi$, (b) $\delta\chi_u$, (c) $\delta T_u$, and (d) $\delta q$. The horizontally averaged standard deviation for $\delta p_{s,u}$ is 53.05 Pa.





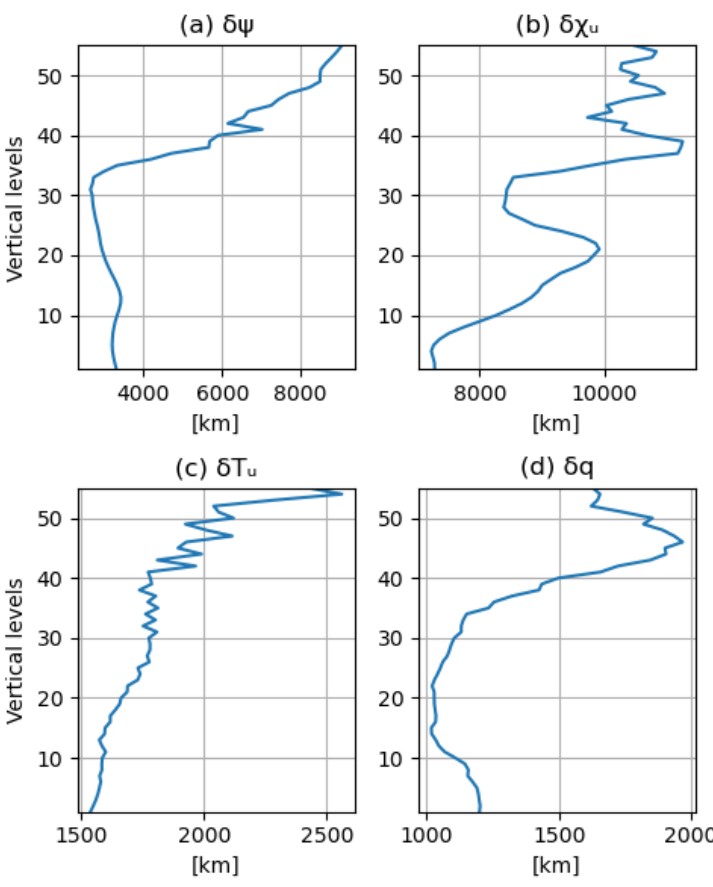

**Figure 4.** Horizontally averaged horizontal correlation lengths [km] for (a) $\delta\psi$, (b) $\delta\chi_u$, (c) $\delta T_u$, and (d) $\delta q$. The horizontally averaged length for $\delta p_{s,u}$ is 3702.3 km.



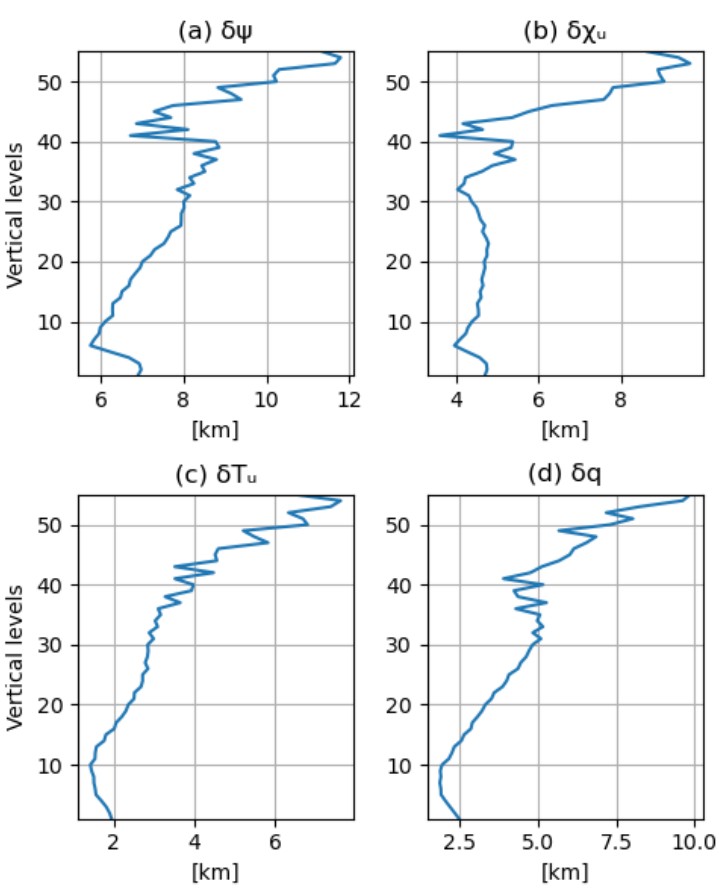

**Figure 5.** Horizontally averaged vertical correlation lengths [km] for (a) $\delta\psi$, (b) $\delta\chi_u$, (c) $\delta T_u$, and (d) $\delta q$.

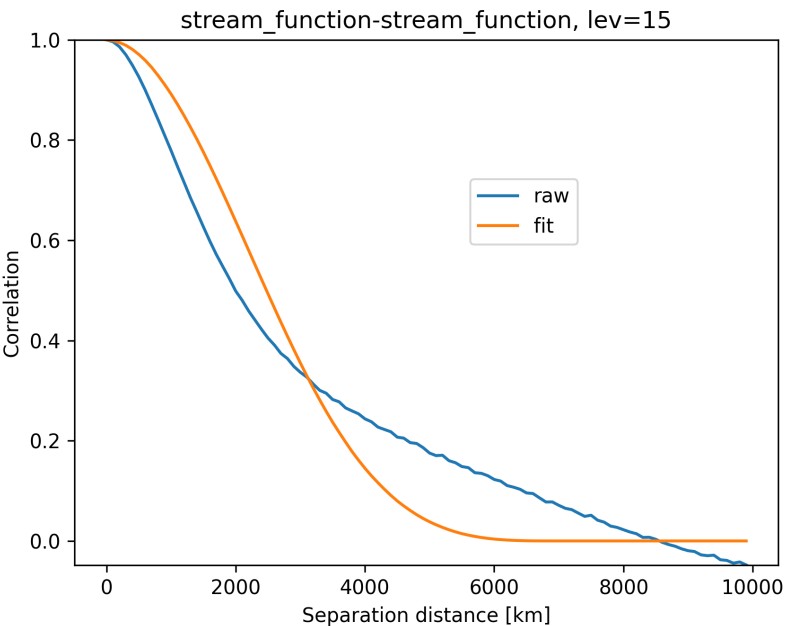

**Figure 6.** The isotropic correlation function for $\delta\psi$ on the 15th model level, based on the sample of forecast differences (blue). Also shown is the correlation function assumed in BUMP (orange) using the length scale the gives the best fit to the sample-derived correlation.



**Figure 7.** Analysis increments for (left column) zonal component of $\boldsymbol{v}$, (center column) meridional component of $\boldsymbol{v}$, and (right column) $T$ on model level (upper row) 10, (middle row) 15, and (lower row) 20, from a single temperature observation with 1 K innovation and 1 K observation-error standard deviation, located at (38.68° W, 40.41° N) on model level 15 with a marker ×.





**Figure 8.** Same as Fig. 7, except from a single zonal wind observation with $1\ \mathrm{ms}^{-1}$ innovation and $1\ \mathrm{ms}^{-1}$ observation-error standard deviation.



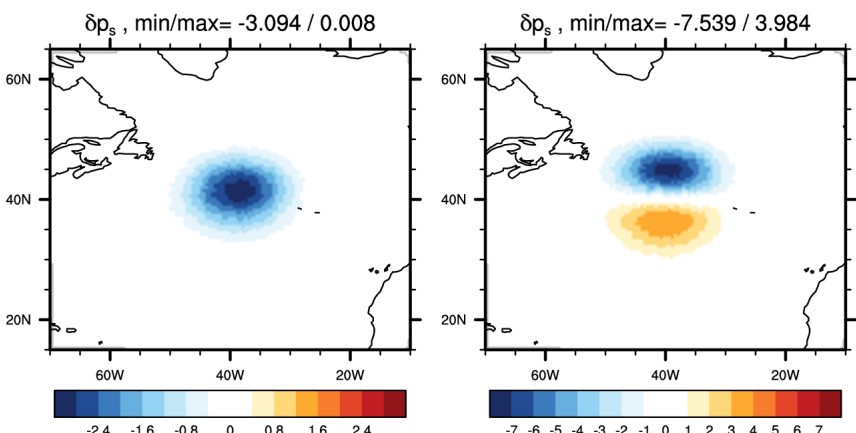

**Figure 9.** Analysis increments for $p_s$ (left) from a single temperature observation with 1 K innovation and 1 K observation-error standard deviation and (right) from a single zonal wind observation with $1\ \mathrm{ms}^{-1}$ innovation and $1\ \mathrm{ms}^{-1}$ observation-error standard deviation





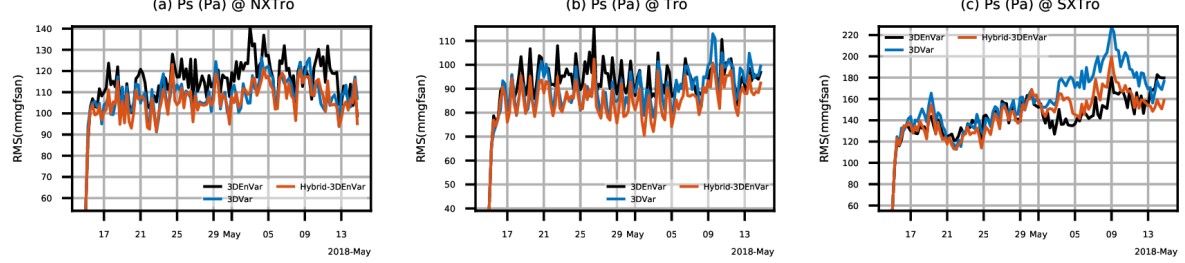

**Figure 10.** Time series (00:00 UTC, 15 April to 18:00 UTC, 14 May 2018) of background RMSEs for $p_s$ verified with GFS analysis over (a) northern extratropical (30–90°N; NXTro), (b) tropical (30°S–30°N; Tro), and (c) southern extratropical (30–90°S; SXTro) region.



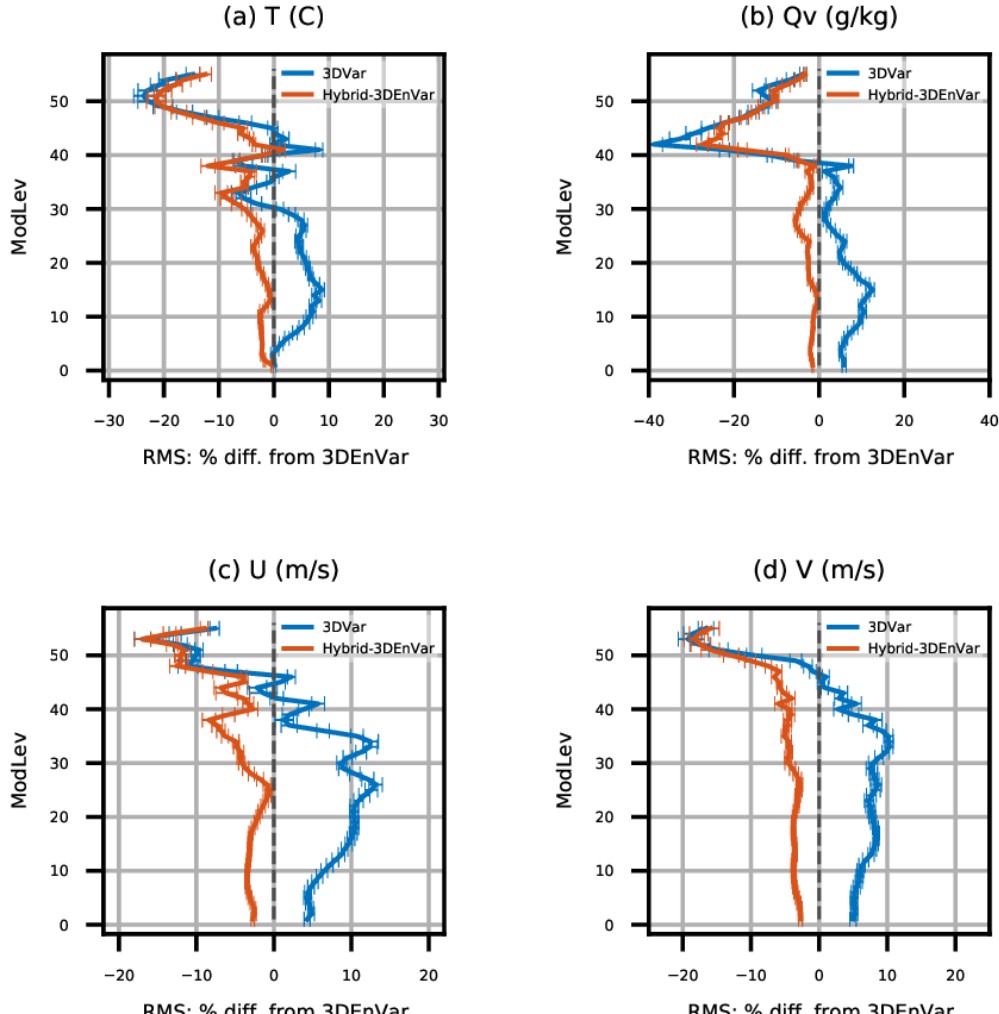

**Figure 11.** Vertical profiles of relative background RMSE changes (with respect to GFS analysis) for (a) zonal wind, (b) meridional wind, (c) $T$, and (d) $q$, compared to 3DEnVar experiment. Statistics are aggregated for the period from 00:00 UTC, 18 April, to 18:00 UTC, 14 May 2018 with 95 % confidence intervals.



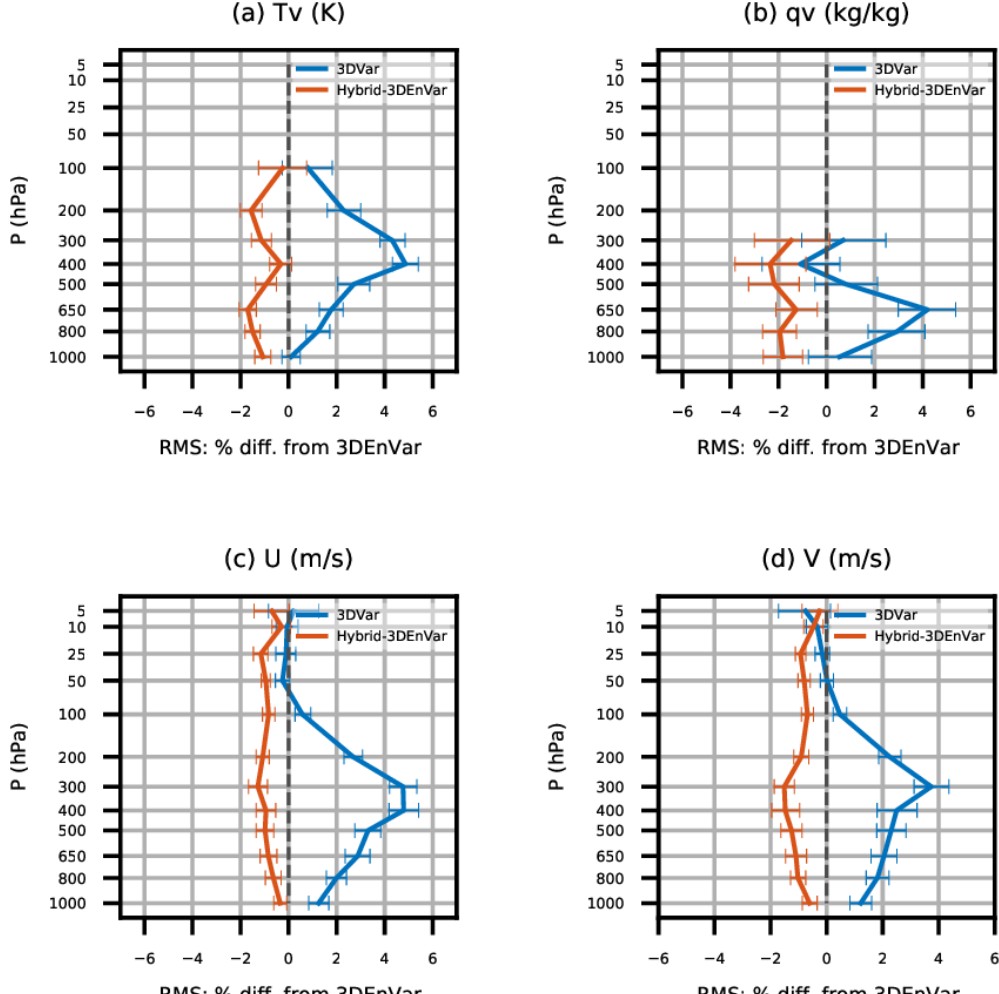

**Figure 12.** Vertical distribution of relative RMS changes of [observation minus background, or first-guess departure] for (a) virtual temperature $T_v$, (b) specific humidity (c) zonal wind, and (d) meridional wind of radiosonde observations. Statistics are aggregated for the period from 00:00 UTC, 18 April, to 18:00 UTC, 14 May 2018 with 95 % confidence intervals.



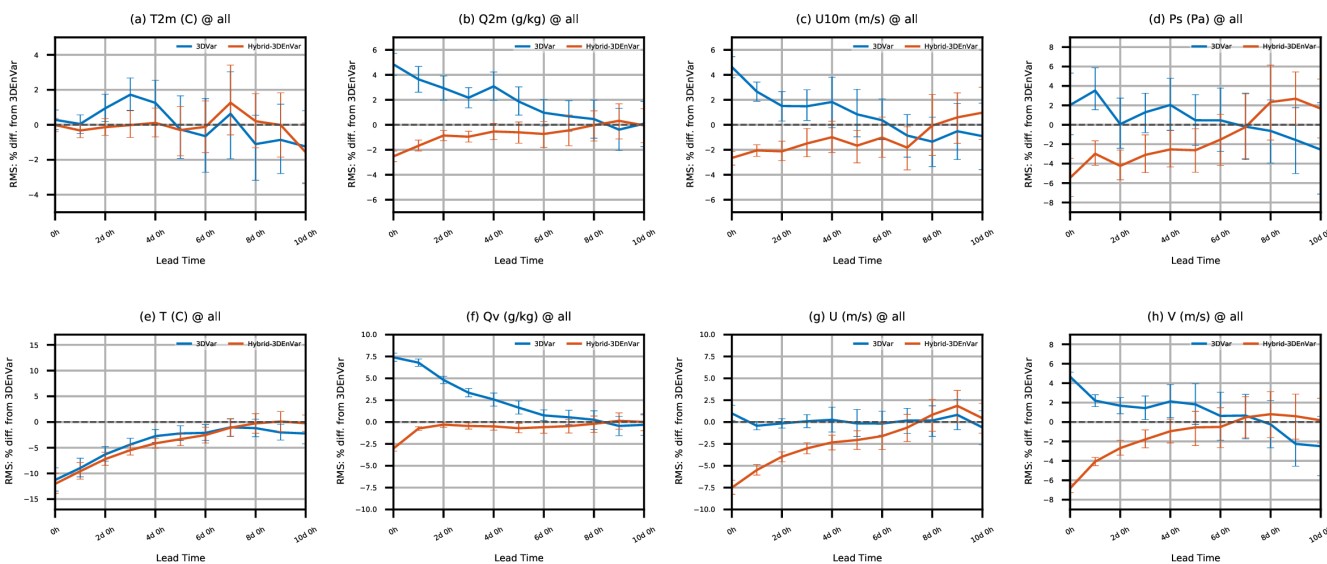

**Figure 13.** Relative RMSE changes as a function of forecast lead time for (upper row) near-surface and (lower row) three-dimensional variables, compared to 3DEnVar experiment. Shown are (a) $T$ at 2 m, (b) $q$ at 2 m, (c) zonal wind at 10 m, (d) $p_s$, (e) $T$, (f) $q$, (g) zonal, and (h) meridional winds. Statistics are aggregated over 27 extended forecasts with 95 % confidence intervals.