# Peer review of "Three-dimensional variational assimilation with a multivariate background error covariance for the Model for Prediction Across Scales–Atmosphere with the Joint Effort for data Assimilation Integration (JEDI-MPAS 2.0.0-beta)"

_Geoscientific Model Development, 2023_

## Author Comment (AC1)

We appreciate the reviewer's detailed comments on the manuscript. Those comments has helped improve and clarify the submitted manuscript. Especially, we have added more detailed information on how BUMP NICAS and HDIAG works. Please find our response to each of your comments below.

1. *Page 1, Line 14: "hybrid covariance that combines that used ..." should be "hybrid covariance, which combines that used..."*

Done.

2. *Page 1, Line 19: missing "centers" after numerical weather prediction*

Done.

3. *Page 2, Lines 44-46: What prevents the products of vectors with the univariate spatial correlation matrices from being computed on the native mesh? It is not clear here as to why it is required to compute in a thinned subset of mesh and interpolated to back to full-resolution mesh. Is this related to BUMP NICAS mentioned later?*

Yes, those lines describes the BUMP NICAS procedure mentioned later. If a single (3-d) variable is in a dimension of [n], the univariate spatial correlation matrices would be in a dimension of [n x n]. Construction of [n x n] matrix and multiplying it with an increment [n] can be too expensive for large systems. Instead, BUMP NICAS constructs the univariate spatial correlation matrix of a subset of mesh [$n_s$ x $n_s$] and interpolation from [$n_s$] to [n], where $n_s$ << n. This reduces the overall computational cost to multiplying the correlation matrix to an increment. Following reviewer's comment and 10th comment, we have revise the last paragraph of section 3.1 to provide more detailed information on how NICAS works.

4. *Page3, Line 56: "heigh-based" should be "height-based"*

Done.

5. *Page 3, Line 67: "the United Forward Operator" should be "the Unified Forward Operator"*

Done.

6. *Page 3, Line 79: "distinguishes" should be "distinguished"*

The "distinguishes" looks correct in the originally submitted manuscript.

7. *In section 2.3, It is a little difficult to understand the new approach (relative to the previous one used in Liu et al. 2022) as described in the second and third paragraphs. The second paragraph talks about analysis variable change, not increments, however, the third paragraph is all about increments even though the variables being described in the two paragraphs are the same. The most unclear part is from line 92 to line 95. How do p, rho_d, and theta_d change relative to background forecast even if no observations were assimilated. Why would background forecast change in this case? And how does this new approach contribute to temperature bias reduction in the stratosphere? Assuming the stratospheric temperature bias was an issue in Liu et al. (2022), please provide more context and consider re-write these two paragraphs for more consistency.*

Thank you for pointing this out.

**[Change from no obs]** In the initial implementation of JEDI-MPAS, the major challenge was on how the (3-d) pressure, $p$, is updated after analyzing surface pressure ($p_s$). The surface pressure, $p_s$, is the analysis variable in JEDI-MPAS, but it is a diagnostic variable in MPAS Model, which is extrapolated from $p$ at the two lowest vertical levels. After minimization is done, the $p$ is computed hydrostatically by integrating (analyzed) T, $p_s$, and q from surface to upper levels, and there happens a non-zero increment for $p$ and consequently for $\rho_d$ and $\theta_d$. This is described as "*discretization errors*" in the original manuscript, and this non-zero increment is accumulated as the vertical indices increase.

**[Benefit of new approach]** By using the incremental formula with base state of DA background, if there is no analysis increment from DA, the background $p$, $\rho_d$, and $\theta_d$ will not change. Also, even with non-zero analysis increment from DA, the error due to the discretization and nonhydrostatic effects will be smaller for $p$, $\rho_d$, and $\theta_d$ by using the incremental formula. We realized that only the "increments" are mentioned in the third paragraph of section 2.3 and this might make the reader confused.

We have revised the first sentence in the third paragraph of section 2.3 as follows: "**Here, we instead compute increments for $p$, $\rho_d$, and $\theta_d$ (i.e., $\delta p$, $\delta\rho_d$, and $\delta\theta_d$) from the increments $\delta T$, $\delta ps$, and $\delta q$, by linearizing the corresponding calculations of Liu et al. (2022, steps 3 and 4 of their section 3.3). Then, the full states of $p$, $\rho_d$, and $\theta_d$ are updated by adding $\delta p$, $\delta\rho_d$, and $\delta\theta_d$ to their background fields. If the analysis increment from DA is zero, in this case, the background state will be kept for $p$, $\rho_d$, and $\theta_d$.**" .

8. *Page 4, Lines 100-101: what is the purpose of mentioning "without a halo region" here? Was it meant to indicate that there are no operations of derivatives or interpolations required for the state(x) and increment (delta x) in JEDI-MPAS as such no halo region is needed? Please explain.*

We'd like to describe technical detail of whether the state and increment objects of JEDI-MPAS contains the halo regions or not. However, the halo exchange should be done when needed, such as horizontal interpolation of state or increment to the observation location, and a linear variation transform from stream function and velocity potential to zonal and meridional winds.

We realized that this might make the readers confusing, so we revised those lines as follows: "**The state and increment objects in JEDI-MPAS only contain their values on own grid point without a halo region. The halo exchange (and its adjoint) is performed when needed, such as horizontal interpolation of state or increment to the observation location and a linear variation transform containing the spatial derivatives.**" in the last paragraph of section 2.3 .

9. *Page 4, Line 106: "of" is missing after "independent"*

Done.

10. *Page 4, Lines 112-113: Please explain in more details on how BUMP models the univariate spatial correlations differently from the recursive filters, which was used in GSI? Perhaps try to make efforts to connect this paragraph with those on the last paragraph (Lines 165-170) of Page 6.*

We have modified the text as "… **except in our use of BUMP-Normalized Interpolated Convolution from on  Adaptive Subgrid (NICAS; Ménétrier, 2020), rather than recursive filters, to model the univariate spatial correlations (see further description at the end of this section).**" and provided more information in the last paragraph of section 3.1 as "**… The spatial correlation matrix is pre-computed from the given correlation lengths with BUMP-NICAS. Similar to the GSI recursive filters, NICAS works in the grid-point space. However, it applies the convolution function explicitly, instead of recursively for GSI. Thus, the choice of the convolution function in NICAS is free, as long as it is positive-definite. We choose a widely-used fifth-order piecewise function of Gaspari and Cohn (1999), which resembles the Gaussian function but is compactly supported. To make the explicit convolution affordable for high-dimensional systems, it is actually performed on a low-resolution unstructured**

**mesh. A linear interpolation is required from the unstructured mesh to the full model grid. Finally, an exact normalization factor is pre-computed and applied to ensure that the whole NICAS correlation operator is normalized (i.e. diagonal elements of the equivalent correlation matrix are "1"). Thus, the NICAS correlation matrix can be written as: $C=NS\widetilde{C}S^{T}N^{T}$, where $\widetilde{C}$ is the convolution operator on the low-resolution mesh, S is the interpolation from the mesh to the full model grid, and N is the diagonal normalization operator. The low-resolution mesh density can be locally adjusted depending on the diagnosed correlation lengths (or provided by the user)."** .

11. *Section 3.2: Another concern I have is regarding the parameters of the multivariate background error covariance being diagnosed from NCEP GFS forecasts, which is a different model from MPAS. The authors mentioned in their future work that they plan to use an ensemble from JEDI-MPAS to train the covariance model, but why not use the MPAS forecasts to diagnose these parameters in the first place? Isn't the MAPS forecast a natural and more straightforward choice that can realistically represent the multivariate structure of the forecast errors of MPAS compared to NCEP GFS? In addition, knowing that the covariance is meant to be used for 6-hourly cycling DA (which is typical), why would the author follow the traditional approach and train the covariance model using data from 24 hour forecast differences and then apply a re-scaling factor to address the gap? This begs the question whether the re-scaling could be avoided if the covariance was trained from 6-h differences of MPAS forecasts. In addition, what is the spatial resolution of the NCEP GFS samples?*

Thank you for your comment. As you mentioned, it is natural to use MPAS Model's own forecast samples to diagnose the B parameters to consider the MPAS Model's own characteristics. In this initial development and validation work, however, we have wanted to use the pre-existing forecast samples from external model. After submission of this manuscript, we did use the MPAS Model's own forecast samples (still with NMC-type perturbations) to diagnose the B parameters, but with a recent version of JEDI-MPAS source code (early June 2023). To summarize, the overall structures of B parameters (such as regression coefficients, vertical profiles of horizontal- and vertical correlation lengths) diagnosed from MPAS-based samples was similar to that from GFS-based samples. The largest difference was in the error standard deviation parameters, which were in larger values for MPAS-based samples, especially in the upper levels. In one-month cycling experiment, this lead to a reduction in temperature and wind RMSEs in 6 hour forecasts fields in the upper levels. In future efforts of further refinement on the JEDI-MPAS static B, we will definitely use the MPAS Model's own forecast samples (either NMC type or ensemble samples).

To generate the sample error perturbations, the NMC method uses two forecast fields with different forecast lead time, verified at the same valid time. The 24 hour different lead time is typical used in defining NMC samples because we want to remove the effect of diurnal

cycle in the perturbations. If we try the ensemble-based samples to diagnose the B statistics in future, it would be natural to use the 6 hour forecast samples, and the rescaling of error standard deviation parameters might be avoided as the reviewer commented.

The resolution of GFS model itself is approximately 13 km (horizontal) with 64 vertical levels. This 13 km GFS model produces the forecast dataset with 0.25 degree lat/lon grids on the standard pressure levels. For GFS-based samples in this study, the 0.25 degree GFS dataset is interpolated into MPAS 60 km grid (also with a vertical interpolation).

We have added: "**Here, the 24-hr forecast lead time difference is chosen to remove the effect of diurnal cycle in the perturbation samples. The GFS forecast files of 0.25 degree resolution on the pressure levels are interpolated to 60 km MPAS mesh with 55 vertical levels for following training procedures.**" in the first paragraph of section 3.2 .

12. *Page 8, Lines 211-219: Does this modification suggest that the assumed correlation function (the fifth-order, compactly supported function from Gaspari and Cohn 1999) is not optimal for stream function and velocity potential, but okay for other variables such as temperature, specific humidity and surface pressure?*

We have found the largest discrepancy between assumed and modelled correlation function for stream function and velocity potential, thus we have aimed to tune the horizontal correlation lengths for these two variables. As described in Lines 215-219, reducing the horizontal correlation lengths for stream function and velocity potential has increased the error variances for zonal and meridional wind analysis variables. This has also greatly improved the fit-to-obs for single zonal wind assimilation test. This does not necessarily means that the use of modelled correlation is okay for other variables, and further efforts are needed to improve the diagnostics of better B parameters.

13. *The authors didn't show analysis increments of specific humidity (q) from the two single observation tests. Is it because there is no correlation between q and other analysis variables (meaning q is univariate) so there is zero q increment? In addition, there is no single observation test for observation of q. Is it due to the same reason?*

Yes, there are no analysis increments from two single observation tests because there is no correlation between specific humidity (q) and other variables. If we perform a single observation test from a single q observation, the increments will be shown only for q fields (i.e., univariate).

This might be the simplest way of handling the moisture analysis variable in the data assimilation. More sophisticated moisture analysis variable and moisture B variables, for

example, pseudo relative humidity (Dee and da Silva, 2003) or normalized relative humidity (Hólm et al, 2002) will be an important future research topic in JEDI-MPAS. We have added this remark at the last paragraph of section 6 as follows: "**We will also explore more sophisticated moisture variables, such as pseudo relative humidity (Dee and da Silva, 2003) or normalized relative humidity (Hólm et al., 2002).**" .

*14. Page 9, Lines 253-257: As stated here, a one-month "cycling" experiments were performed, but at "each" cycle, a 20-member ensemble of 6-hour MPAS forecast was performed using IC from GEFS. Are the experiments fully cycled or partial cycled? The latter one means each cycle always cold-start from GEFS. I would think the experiments are fully cycled, but it is not clear to me which approach was really taken by the authors.*

Yes, the experiments are fully cycled for the *deterministic* analysis. However, the 6-hour MPAS ensemble forecasts were performed as a cold-start mode, initialized from GEFS analysis ensemble. Then, the 6-hour MPAS ensemble forecasts have provided the ensemble background covariance at the valid time.

We have added a text "… **to provide the ensemble background error covariance**."

*15. As stated in the conclusions, the formulation of the JEDI-MPAS static B generally follows Wu et al. (2002), but with the novel use of BUMP for multiple elements of the covariance model. Although the BUMP package including VBAL and VAR are introduced to perform variable transforms, these transformations generally follow Wu et al. (2002). The most novel parts of BUMP are perhaps the NICAS and HDIAG drivers that are used to model the univariate correlation in place of recursive filters. The NICAS and HDIAG drivers allow one to specify spatial correlation functions and compute convolutions on (semi-) native mesh, which should be considered an enhancement over recursive filters used in GSI. However, the subsequent modification which halves the diagnosed horizontal correlation length for stream function and velocity potential makes this enhancement less promising. If the specified correlation function (i.e., the fifth-order compactly supported function from GC1999 chosen in this study) leads to results that are far from the sample statistics, then why not make efforts to find a more appropriate correlation function that may be more suitable for certain variables? It gives the impression that the novelty that BUMP brought about was not fully exploited.*

One biggest benefit of BUMP NICAS/HDIAG is that it can be applied on an unstructured grid, while other spatial correlation operators, for example, recursive filters (in GSI or WRF DA) or spectral transforms in (SSI or UKMO), require a structured grid, such as (reduced) gaussian grid or orthogonal grid with two directions. The GC1999 function models the compactly supported Gaussian shape with a *single* parameter (i.e., separation distance

from the origin), so it is very efficient both when we diagnose the correlation lengths and when we calculate the convolution.

While choosing the other functions can be an interesting research topic, now NICAS has a more advanced capability to diagnose and to apply the multiple components of GC functions. By superposing the multiple GC functions with different correlation lengths, this capability is expected to fit the raw sample correlation better. This has not yet been tested in a large dataset, and we'd like to test this with JEDI-MPAS in near future and will see if it resolves the issue of additional tuning. This was briefly mention at the last sentence of section 6 in the original manuscript.

16. *Figure 1 Caption: regression coefficients of (a) delta T, (b) delta phi, and (c) delta kai are the nonzero elements at this mesh cell of the submatrices should correspond to M, L, and N, respectively..., not L, M and N, according to Equation (4).*

Thank you for finding this. It is fixed.

17. *Figure 6.: "using the length scale the gives the best fit" should be "using the length scale that gives the best fit"*

Done.

18. *Figure 7: Please draw the reduced horizontal correlation length in a different color to show the effect of the additional modifications made to the raw statistics.*

Done.

19. *Figures 7-9: Please include variable units.*

Variable units are added in the color bars and figure captions.

**Reference**

Dee, D. P. and da Silva, A. M.: The Choice of Variable for Atmospheric Moisture Analysis, Monthly Weather Review, 131, 155 – 171, https://doi.org/https://doi.org/10.1175/1520-0493(2003)131<0155:TCOVFA>2.0.CO;2, 2003.

Gaspari, G. and Cohn, S. E.: Construction of correlation functions in two and three dimensions, Quarterly Journal of the Royal Meteorological Society, 125, 723–757, https://doi.org/https://doi.org/10.1002/qj.49712555417, 1999.

Hólm, E., Andersson, E., Beljaars, A., Lopez, P., Mahfouf, J.-F., Simmons, A., and Thépaut, J.-N.: Assimilation and Modelling of the Hydrological Cycle: ECMWF's Status and Plans, ECMWF Technical Memoranda, p. 55, https://doi.org/10.21957/kry8prwuq, 2002.

Liu, Z., Snyder, C., Guerrette, J. J., Jung, B.-J., Ban, J., Vahl, S., Wu, Y., Trémolet, Y., Auligné, T., Ménétrier, B., Shlyaeva, A., Herbener, S., Liu, E., Holdaway, D., and Johnson, B. T.: Data assimilation for the Model for Prediction Across Scales – Atmosphere with the Joint Effort for Data assimilation Integration (JEDI-MPAS 1.0.0): EnVar implementation and evaluation, Geoscientific Model Development, 15, 7859–7878, https://doi.org/10.5194/gmd-15-7859-2022, 2022.

Ménétrier, B.: Normalized Interpolated Convolution from an Adaptive Subgrid documentation, https://github.com/benjaminmenetrier/nicas_doc/blob/master/nicas_doc.pdf, 2020.

---

## Author Comment (AC2)

We appreciate the reviewer's detailed comments on the manuscript. Those comments has helped improve and clarify the submitted manuscript. Especially, we have added more detailed information on how BUMP NICAS and HDIAG works. Please find our response to each of your comments below.

1. *Line 56: "heigh" -> "height"*

Done.

2. *Line 67: "United" -> "Unified"*

Done.

3. *Line 132-133: Can you clarify the purpose of using the same level only for calculating regression coefficients for δψ and δχ? Do you think their vertical cross-correlations are weak/negligible?*

It could be possible to have vertical cross-correlations between *δψ and δχ.* This could be tested and evaluated if that explains bigger amount of total sample variance (shown in Figure 2b). In this manuscript, however, we have considered the level-by-level correlation between *δψ and δχ*, following Wu et al. (2002).

We have added a text "**..., following Wu et al. (2002).**"

4. *Line 138: I think this manuscript can contribute much to the research community. If the authors can include how BUMP implements these operators from the algorithm perspective in detail, this manuscript can be at a higher level.*

Thank you for the suggestion. We have improved the description on how BUMP VBAL, VAR, and especially NICAS works in the last paragraph of section 3.1 as follows: "**The BUMP Vertical BALance (VBAL) driver is used for $K_2$ and $K_2^T$ . It is based on the explicit vertical covariance matrices defined for a set of latitudes, and interpolated at the model grid points latitude. The BUMP VARiance (VAR) driver is used for $\Sigma$ and $\Sigma^T$ . It simply applies the pre-computed error standard deviations. The spatial correlation matrix is pre-computed from the given correlation lengths with BUMP-NICAS. Similar to the GSI recursive filters, NICAS works in the grid-point space. However, it applies the convolution function explicitly, instead of recursively for GSI. Thus, the choice of the convolution function in NICAS is free, as long as it is positive-definite. We choose**

a widely-used fifth-order piecewise function of Gaspari and Cohn (1999), which resembles the Gaussian function but is compactly supported. To make the explicit convolution affordable for high-dimensional systems, it is actually performed on a low-resolution unstructured mesh. A linear interpolation is required from the unstructured mesh to the full model grid. Finally, an exact normalization factor is pre-computed and applied to ensure that the whole NICAS correlation operator is normalized (i.e. diagonal elements of the equivalent correlation matrix are "1"). Thus, the NICAS correlation matrix can be written as: $C = NS\tilde{C}S^{T}N^{T}$, where $\tilde{C}$ is the convolution operator on the low-resolution mesh, S is the interpolation from the mesh to the full model grid, and N is the diagonal normalization operator. The low-resolution mesh density can be locally adjusted depending on the diagnosed correlation lengths (or provided by the user)."

Also, we have added more description on BUMP VAR and especially HDIAG in the last two paragraph of section 3.2 as follows: "**For $\Sigma$, the BUMP VAR driver calculates variances for $\delta\psi$, $\delta\chi_u$, $\delta T_u$, $\delta q$, and $\delta p_{s,u}$ from the samples and filters them horizontally to damp the sampling noise. The horizontal smoother is also based on NICAS, with an appropriate mean-preserving normalization factor.**

**The correlation matrix, C, consists of blocks that specify the univariate spatial correlation for $\delta\psi$, $\delta\chi_u$, $\delta T_u$, $\delta q$, and $\delta p_{s,u}$. The BUMP Hybrid DIAGnostic (HDIAG) driver diagnoses the horizontal and vertical correlation lengths used in modeling C parameters. HDIAG can diagnose the horizontal and vertical spatial correlations from the samples. First, it defines a low- resolution unstructured mesh. Around each mesh node, diagnostic points are randomly and isotropically drawn for different horizontal separation classes. Second, HDIAG calculates the horizontal correlation between each mesh node and its own diagnostic points from the samples, at all levels. The vertical correlation is also calculated at each mesh node, between each level and the neighboring levels. The third step is a horizontal averaging of these raw correlations over all the mesh nodes. The average is binned depending on the level and the horizontal separation for the horizontal correlation, and depending on the concerned levels for the vertical correlation. As a final step, HDIAG fits a Gaspari and Cohn (1999) function for each averaged correlation curve. Thus, we obtain horizontal and vertical correlation length values for each level. These profiles can be stored and provided to NICAS in order to model the spatial correlation operator."**

5. *Line 147: You directly used GFS forecasts to calculate the static error statistics. Are GFS forecasts appropriate to be used to represent MPAS model errors? I doubt it.*

Thank you for your comment. As you mentioned, it is natural to use MPAS Model's own forecast samples to diagnose the B parameters to consider the MPAS Model's own characteristics. In this initial development and validation work, however, we have wanted to

use the pre-existing forecast samples from external model. After submission of this manuscript, we did use the MPAS Model's own forecast samples (still with NMC-type perturbations) to diagnose the B parameters, but with a recent version of JEDI-MPAS source code (early June 2023). To summarize, the overall structures of B parameters (such as regression coefficients, vertical profiles of horizontal- and vertical correlation lengths) diagnosed from MPAS-based samples was similar to that from GFS-based samples. The largest difference was in the error standard deviation parameters, which were in larger values for MPAS-based samples, especially in the upper levels. In one-month cycling experiment, this lead to a reduction in temperature and wind RMSEs in 6 hour forecasts fields in the upper levels. In future efforts of further refinement on the JEDI-MPAS static B, we will definitely use the MPAS Model's own forecast samples (either NMC or ensemble types of samples).

6. *Line 152: Using NCL first seems to make the procedure complicated. Is it an essential step, or the alternative strategies exist?*

Unfortunately, it is essential step for current strategy and yes, this makes the B training procedure a bit complicated. As described in the manuscript, it is not trivial to solving a Poisson equation efficiently on the unstructured grid. However, this step is only required once when we generate the perturbation samples for stream function and velocity potential from zonal and meridional wind.

7. *Line 166-171: This is one novelty part relative to the other utilities (e.g., gen_be_v2 in GSI), right? I suggest the authors give more details about HDIAG.*

We have added more description on BUMP HDIAG in the last paragraph of section 3.2 . Please see the response on the reviewer's 4[th] comment above.

8. *Figure 8: Is it the same observation location used as Fig.7, but for the zonal wind? Can you clearly state the location of the single zonal wind observation and mark it in Fig. 8?*

Yes, the location is the same for zonal wind observation and temperature observation. It is marked as "x" in the left-mid panel. We also added the "x" mark in the other panels and revised the Figure 8 caption as follows: "**Same as Fig. 7, except from a single zonal wind observation with 1 ms⁻¹ innovation and 1 ms⁻¹ observation-error standard deviation, located at (38.68° W, 40.41° N) on model level 15 with a marker ×.**"

9. *Figure 9: Same as the comment for Fig. 8. Please mark the location of the assimilated observations.*

The observation location is marked with "x" in the revised figure.

**Reference**

Gaspari, G. and Cohn, S. E.: Construction of correlation functions in two and three dimensions, Quarterly Journal of the Royal Meteorological Society, 125, 723–757, https://doi.org/https://doi.org/10.1002/qj.49712555417, 1999.

Wu, W.-S., Purser, R. J., and Parrish, D. F.: Three-dimensional variational analysis with spatially inhomogeneous covariances, Monthly Weather Review, 130, 2905–2916, https://doi.org/10.1175/1520-0493(2002)130<2905:TDVAWS>2.0.CO;2, 2002.

---

## Author Response (AR2)

**[ For Topical Editor ]**  We appreciate the topical editor's comments on the manuscript. Please find our response to each of your comments below.

- *I agree with the reviewers' suggestion about elaborating on the implementation and use of BUMP, which is an essential contribution to this work.*

We have made additional revisions as described in our responses to reviewer 1, comment 3, and reviewer 2, comments a and b.

- *The authors should briefly explain or show a figure to quantitatively discuss how tuning the horizontal lengths of \psi and \chi by half changes the velocity variance.*

We have changed lines **246-252** to clarify how the implied velocity variance changes with changes in the $\delta\psi$ and $\delta\chi_u$ correlation lengths. (Since the velocity variance is proportional to the second derivative at the origin of the $\delta\psi$ and $\delta\chi_u$ correlation function, the velocity variance is inversely proportional to the square of the correlation length.)

**Previous text:**
Since the implied velocity variance depends on the second derivative at the origin of the $\delta\psi$ (and $\delta\chi$) correlation (Lorenc, 1981; Daley, 1985), the diagnosed covariances greatly underestimate the velocity variance relative to the statistics of the of original training data. Reducing the horizontal correlation length for $\delta\psi$ and $\delta\chi_u$ increases the velocity variances, though at the expense of further underestimating the correlations at larger separations.

**New text:**
The implied velocity variance in the modeled covariance (Eq. 2) is proportional to the second derivative at the origin of the $\delta\psi$ (and $\delta\chi$) correlation (Lorenc, 1981; Daley, 1985). That is, $\delta\psi$ correlations that are more strongly peaked at the origin will produce larger velocity variance even if the $\delta\psi$ variance is fixed.  Thus, the modeled covariances greatly underestimate the velocity variance relative to the statistics of the original training data. Reducing the horizontal correlation length for $\delta\psi$ and $\delta\chi_u$ by a factor of two increases the second derivative of their correlation, and therefore the velocity variance, by a factor of 4, leading to a better fit to the velocity variance in the training data.

- *Also, please address the issue of the poorer performance of the 3DVar Qv analysis compared to 3DEnVar (Fig. 13). This may explain why using sophisticated moisture variables can help.*

Thank you for your suggestions. We have added the following sentences in lines 327-328 (section 5.2):
"**It is notable that the larger Qv RMSE for 3DVar lasts until 6 day forecasts (Fig. 14f). This might be because the moisture variable is univariate in current B design (section 3.1) together with relatively less observation amount for moisture.**" . Two literatures for moisture variables are mentioned in section 6, so they are not mentioned here.

**[ For reviewer 1]**  We appreciate the reviewer's comments on the manuscript. Please find our response to each of your comments below.

> 1. *Section 2.3: I very much appreciate the author's detailed responses to my previous comments regarding the background of this new incremental approach for updating p, roh_d, and theta_d. Nevertheless, I think revisions should be extended to the second paragraph as well. It is important to point out that the 3-dimensional pressure p is a prognostic variable in MPAS but not being used as an analysis variable in JEDI here in this manuscript. As such, after each data assimilation analysis, pressure is "re-diagnosed" from integrating T, q, and p_s from surface to upper levels via hydrostatic balance, resulting in a pressure that is different from the background forecast (i.e., a non-hydrostatic pressure). This discretization errors can exist even if there is no analysis increment from DA. The way the second paragraph is written does not state this very clearly, although it was very clearly explained in the author's response and should be included in the manuscript.*
> *However, after reading the author's responses, I have another question. If there is no analysis increment from data assimilation, why not just use the background file as the analysis file, why is there a need to propose a specific treatment to this zero-increment situation? Or perhaps when you say "increment from DA is zero" you meant zero increment for ps only while increments for other analysis variables are non-zero, as such, you can't just replace analysis file with background file.*

We have revised both paragraphs, considering the reviewer's comments from both rounds of reviews. We now emphasize the central point that Liu et al. (2022) enforced hydrostatic balance on the full, analyzed fields, while the new approach used in this paper applies hydrostatic balance only to increments from the background fields.  We have also omitted some confusing details, such as the role of pressure as an intermediary variable in the transformations (note that pressure is not a prognostic variable in MPAS), and the fact that the incremental approach avoids any changes to the background state in regions where there are no observations.

Please see the revised paragraphs at lines 93-102:
"**In Liu et al. (2022), $\rho_d$ and $\theta_d$ are computed from the analyzed T, $p_s$, and q by assuming hydrostatic balance. Here, we instead compute increments for $\rho_d$, and $\theta_d$ (i.e., $\delta\rho_d$, and $\delta\theta_d$) from the increments $\delta T$, $\delta p_s$, and $\delta q$. This approach, which is implemented by linearizing the corresponding calculations of Liu et al. (2022, steps 3 and 4 of their section 3.3), assumes hydrostatic balance only for the increments and not the full, analyzed fields.**

   **Assuming hydrostatic balance just for the increments is preferable because that balance is only approximate and, moreover, the discretized form of hydrostatic balance used in the variable transformation is not precisely equivalent to that implied by the discrete MPAS equations. Since the hydrostatic integral is computed from the surface upward, differences between the incremental and full-fields formulations can be expected to accumulate with height. Consistent with this, JEDI-MPAS cycling experiments (not shown) using the new, incremental update for $\rho_d$ and $\theta_d$ exhibit reduced temperature bias in the stratosphere, especially near the model top.**" .

2. *Section 3.2: Regarding the training dataset, the author stated in their response that they actually have used the MPAS model's own forecast samples to diagnose the B parameters and compared with those from the NCEP GFS forecast samples and found overall similarity except for the error standard deviation having larger differences. Furthermore, the resulting one-month cycling experiments shown reduced temperature and wind RMSE in the upper levels for the 6-h forecasts when MPAS model's own forecast samples were used to diagnose B parameters. I think this is worth mentioning in the revised manuscript even though it is from a recent version of JEDI-MPAS source code that is different from the one used here.*

Thank you for the reviewer's suggestion. We have added the following statement to the end of the first paragraph in section 3.2 (lines 166-170).
"**With a recent (early June 2023) version of JEDI-MPAS source code after initial submission of this paper, we have trained the static B parameters from MPAS model's own forecast with the same methodology described here. The overall B statistics diagnosed from MPAS-based samples were similar to that from GFS-based samples reported here, except for the error standard deviations in the stratosphere, which were larger for MPAS-based samples. In the one-month cycling experiment, this led to a reduction in temperature and wind RMSEs in 6 hour forecasts in the stratosphere.**" .

3. *I agree with the other reviewer's comment that this manuscript can be at a higher level by considering to address the BUMP implementation in more details, especially from the algorithmic perspective. The authors have taken the suggestions and provided more details in sections 3.1 and 3.2 of the revised manuscript, which is very nice. I think this revised manuscript can be further elevated by including a flowchart or diagram to illustrate the BUMP's implementation, highlighting the correspondence between each of the operator in equation (2) with its associated BUMP driver(s). For example, C, the block-diagonal correlation matrix in equation (2) involves the use of NICAS and HDIAG, while Σ, the diagonal matrix of standard deviations involves the use of VAR and NICAS. Including a flowchart or diagram will also make this manuscript more educational and attract more readers.*

Thank you for your suggestion. We have added a simple diagram to represent the operation shown in equation (2) as figure 1. The following text was added at the end of section 3.1 (lines 158-159) :
"**Figure 1 shows a diagram for Eq. 2 with corresponding BUMP drivers and MPAS-specific linear variable change.**" .

**[ For reviewer 2]** We appreciate the reviewer's comments on the manuscript. Please find our response to each of your comments below.

    *a)*   *At the third step of HDIAG, a horizontal average is performed over all mesh nodes. I wonder if it is possible to average parts of mesh nodes to achieve spatially varying statistics.*

Indeed, HDIAG can generate the *local* statistics, but we missed this in the steps of HDIAG. We have changed lines 195-202 as follows:

"The third step is a horizontal averaging of these raw correlations**, either over all the mesh nodes or over local neighborhoods**. The average is binned depending on the level and the horizontal separation for the horizontal correlation, and depending on the concerned levels for the vertical correlation. As a final step, HDIAG fits a Gaspari and Cohn (1999) function for each averaged correlation curve. Thus, we obtain horizontal and vertical correlation length-scale values for each level. **If the averaging and curve fitting steps are performed over local neighborhoods, an extra interpolation step is necessary to obtain 3D fields of length-scales on the model grid.** These **length-scale profiles or 3D fields** can be stored and provided to NICAS in order to model the spatial correlation operator. **In this study, the local correlation lengths were obtained from raw statistics within 3000 km radius for a given diagnostic point.**" .

    *b)*   *How is the cost of HDIAG compared to the preexisting methods in calculating these correlation statistics?*

Any method needs to diagnose some statistics (sample correlation) from samples. BUMP HDIAG does this on the subsampled grid (which is beneficial in a computational aspect), then interpolate those back to the full grid (this requires an additional cost, but marginal compared to benefit from subsampling).

---

## Author Response (AR3)

**[ For Topical Editor ]**  We appreciate the topical editor's comments on the manuscript. Please find our response to each of your comments below.

> *1. Please clarify "relatively less observation amount for moisture". Did the authors mean to compare the amount of moisture observations to other variables, such as wind?*

We realized that the statement is misleading. Originally, we were thinking of "effective" number of observation, which can affect the moisture analysis. Compared to the ensemble covariances, "relatively" less amount of observation may contribute the analysis of moisture variate in the univariate static covariance.

To make this more clear, we revise the text as…  "This might be because the moisture variable is univariate in current B design (section 3.1), **while the moisture analysis in 3DEnVar or Hybrid-3DEnVar can be done through the multivariate ensemble covariances."**

> *2. I suggest that address the fact that the moisture variable is univariate in current B design may limit the performance of moisture analysis even with hybrid-3DVAR in the conclusion section (line 375).*

We added the text as follows in a given line: "**In current B design, the moisture variable is univariate that may limit the performance of moisture analysis even with Hybrid-3DEnVar configuration. To this end,** we will explore …"